# Effect of holiday admission for acute aortic dissection on in-hospital mortality in Japan: A nationwide study

**Katsuhito Kato**[1]*, **Toshiaki Otsuka**[1], **Michikazu Nakai**[2], **Yoko Sumita**[2], **Yoshihiko Seino**[3], **Tomoyuki Kawada**[1]

**1** Department of Hygiene and Public Health, Nippon Medical School, Bunkyo-Ko, Tokyo, Japan, **2** Centre for Cerebral and Cardiovascular Disease Information, National Cerebral and Cardiovascular Center, Osaka, Japan, **3** Cardiovascular Centre, Nippon Medical School Chiba-Hokusoh Hospital, Chiba, Japan

* katzkato@nms.ac.jp

**Data Availability Statement:** Data cannot be shared publicly. In order to handle the data, you must be a member of the Japanese Society of Cardiology (JSC) according to the The Japanese

## Abstract

### Background

Patients admitted on weekends have higher mortality than those admitted on weekdays. However, whether the "weekend effect" results in a higher mortality after admission for acute aortic dissection (AAD),—classified according to Stanford types—remains unclear. This study aimed to examine the association between admission day and in-hospital mortality in AAD Type A and B.

### Methods

We used data from the Japanese registry of all Cardiac and Vascular Diseases Diagnostic Procedure Combination, a nationwide claim-based database with data from 953 certified hospitals, and enrolled in-patients with AAD admitted between April 1, 2012, and March 31, 2016. Based on the admission day, we stratified patients into groups (Weekdays, Saturdays, and Sundays/holidays). The influence of the admission day on in-hospital mortality was assessed via multi-level logistic regression analysis. We also performed a Stanford type-based stratified analysis.

### Results

Among the included 25,641 patients, in-hospital mortality was 16.0%. The prevalence of patients admitted with AAD was relatively higher on weekdays. After adjustment for covariates, patients admitted on a Sunday/holiday showed an increased risk of in-hospital mortality (odds ratio [OR] 1.20; 95% confidence interval [CI] 1.07–1.33, p<0.001) than patients admitted on weekdays. Among patients admitted on a Sunday/holiday, only the subgroup of Stanford Type A showed a significantly increased risk of in-hospital mortality. (Stanford Type A, non-surgery vs. surgery groups: 95% CI 1.06–1.48 vs. 1.17–1.68, p<0.001 for both groups, OR 1.25 vs. 1.41, respectively, Stanford Type B, non-surgery vs. surgery groups: 95% CI 0.64–1.09 vs. 0.40–2.10; p = 0.182 vs. 0.846; OR 0.84 vs. 0.92).

Registry Of All cardiac and vascular Diseases (JROAD) regulations operated by the JSC. Data are available from the JROAD (contact via ITdatabase@j-circ.or.jp) for researchers who meet the criteria for access to confidential data.

**Funding:** KK The present work was supported by a grant from JSPS KAKENHI grant number 19K19470. https://www.jsps.go.jp/j-grantsinaid/ The funders had no role in study design, data collection and analysis, decision to publish, or preparation of the manuscript.

**Competing interests:** The authors have declared that no competing interests exist.

## Conclusions

In conclusion, patients with AAD Type A admitted on a Sunday/holiday may have an increased in-hospital mortality risk.

## Introduction

Acute aortic dissection (AAD), a medical emergency, is a highly lethal condition. Its incidence rate is lower than that of other cardiovascular diseases, such as acute coronary syndrome; therefore, epidemiological studies regarding AAD remain insufficient [1].

The association between weekend admissions and an increased risk of mortality, the "weekend effect" has received much attention over the last two decades [2]. The findings of studies on the "weekend effect" included various acute phase disease management measures such as emergency and intensive care admissions, as well as urgent surgeries; however, none of the studies attributed these measures to the "weekend effect" [2–5]. The "weekend effect" seems to be influenced by the characteristics of the disease or medical systems [6]. Under such circumstances, only a few studies have reported associations between weekend admissions and mortality in AAD [7–9], and the findings regarding the "weekend effect" in AAD were also inconsistent. Kumar et al. [8] and Gallerani et al. [7], in cohort studies in the United States and Italy, respectively, have reported that admission for AAD on a weekend is associated with a significantly higher mortality rate than admission on a weekday. In contrast, Gokalp et al. reported that there was no statistically significant difference between patients with Type A AAD who underwent surgery during daytime working hours and those who underwent surgery during overtime hours in Turkey [10]. Furthermore, Ahlsson et al. reported that while there was no "weekend effect," nighttime surgery was a risk factor for mortality according to an analysis of European registry data [9]. Moreover, no nationwide studies have been reported the weekend effect in patients with AAD classified by Stanford type, although the natural history, complications, and treatment of AAD differ between Stanford Type A and B patients [11].

Therefore, using the nationwide claim-based database from The Japanese Registry of All Cardiac and Vascular Diseases Diagnosis Procedure Combination (JROAD-DPC), we investigated associations between day-wise variations in admissions for AAD and patient prognosis in a large cohort.

## Methods

### Study setting and patient selection

This study analyzed data between April 2012 and March 2016, from the JROAD-DPC database, a nationwide, claim-based registry from nearly all teaching Japanese hospitals with beds reserved for cardiovascular patients. The JROAD-DPC database has been described in detail previously [12, 13], and can be briefly described as follows: the JROAD-DPC database established by the Japanese Circulation Society comprises data from teaching hospitals with cardiovascular beds and the Japanese DPC/Per Diem Payment System. This database contain patients' demographic and disease-specific data. The validity of the diagnoses contained therein has been reported [14].

Using the International Classification of Disease (ICD)-10 diagnosis codes for AAD (I70-0) as a means of identification, the data of adult patients (age >18 years) who were admitted to

the hospital with AAD and patients who were admitted for emergencies were extracted from the JROAD-DPC. In addition, the diagnosis, recorded in Japanese, was used to increase the accuracy of the diagnoses; for example, patients with ruptured aortic aneurysms, secondary to trauma, sub-acute or chronic status of AAD, or dissection of peripheral arteries suspicion were excluded. The type of surgery and whether the presence or absence of surgery was related to AAD, recorded in Japanese, were also used to increase accuracy. Using the diagnosis, recorded in Japanese, the Stanford type was determined. A Stanford Type A dissection was defined as any dissection that involved the ascending aorta, whereas a Type B dissection was defined as one that involved the descending aorta. All patients were categorized into seven 1-day-of the week periods. In Japan, we have approximately 17 days of non-weekend holidays annually, considered as national holidays. For the weekday analyses, patients were grouped according to their day of admission into the following groups: weekday (Monday to Friday), Saturdays, and Sunday/holiday groups, and a group for each of the seven days. Saturdays and the Sunday/holiday groups were considered as separate groups, given that not all Japanese hospitals closed on Saturdays.

Patients' hospitalization dates included the same days between 0:00 am and 11:59 pm.

Between April 2012 and March 2016, a total of 3,626,656 patients were registered in the database. According to the specified ICD-10 codes, 51,608 patients were diagnosed with AAD. The data of 25,641 patients were finally analyzed in this study. The flowchart of the patient selection process is shown in Fig 1.

## Ethical considerations

Our study protocol was approved by the Ethics Committee of the Nippon Medical School, Japan (approval number 30–03). The requirement for written informed consent was waived because of the anonymized nature of the data.

## Measurement

The primary outcome was in-hospital mortality. The secondary outcome was in-hospital mortality by Stanford type. Our database contained data of patients who were brought to the hospital for cardiac arrest and were diagnosed with AAD during their hospital stay. The death of such patients was considered as in-hospital mortality. Contrastingly, the death of patients who were in a dying state and discharged before death upon family request was not considered as in-hospital mortality.

## Statistical analysis

Patient characteristics and outcomes in terms of the patients' sex and Stanford type were expressed as means ± standard deviations (SD), medians (interquartile range), or numbers (percentage of total), as appropriate. The correlation between baseline characteristics and each group was compared using one-way analysis of variance (ANOVA), the Kruskal–Wallis test, or $X^2$ test, as appropriate. Regarding the frequency analysis, the average number of patients admitted per day was compared among the groups (weekdays vs. Saturdays vs. Sundays/holidays and each of the seven- day groups) using ANOVA with post-hoc Bonferroni correction. The influence of the day of admission on in-hospital mortality was assessed using a multi-level logistic regression analysis with the institute as a random intercept adjusting for possible confounders such as sex, age, Stanford type, and surgery. We compared the day of admission among weeks (weekdays [used as a reference], Saturdays vs. Sundays/holidays and groups for each of the seven days [Sundays/holidays was used as a reference]). We also performed a stratified analysis according to the Stanford type and presence or absence of surgery. Statistical

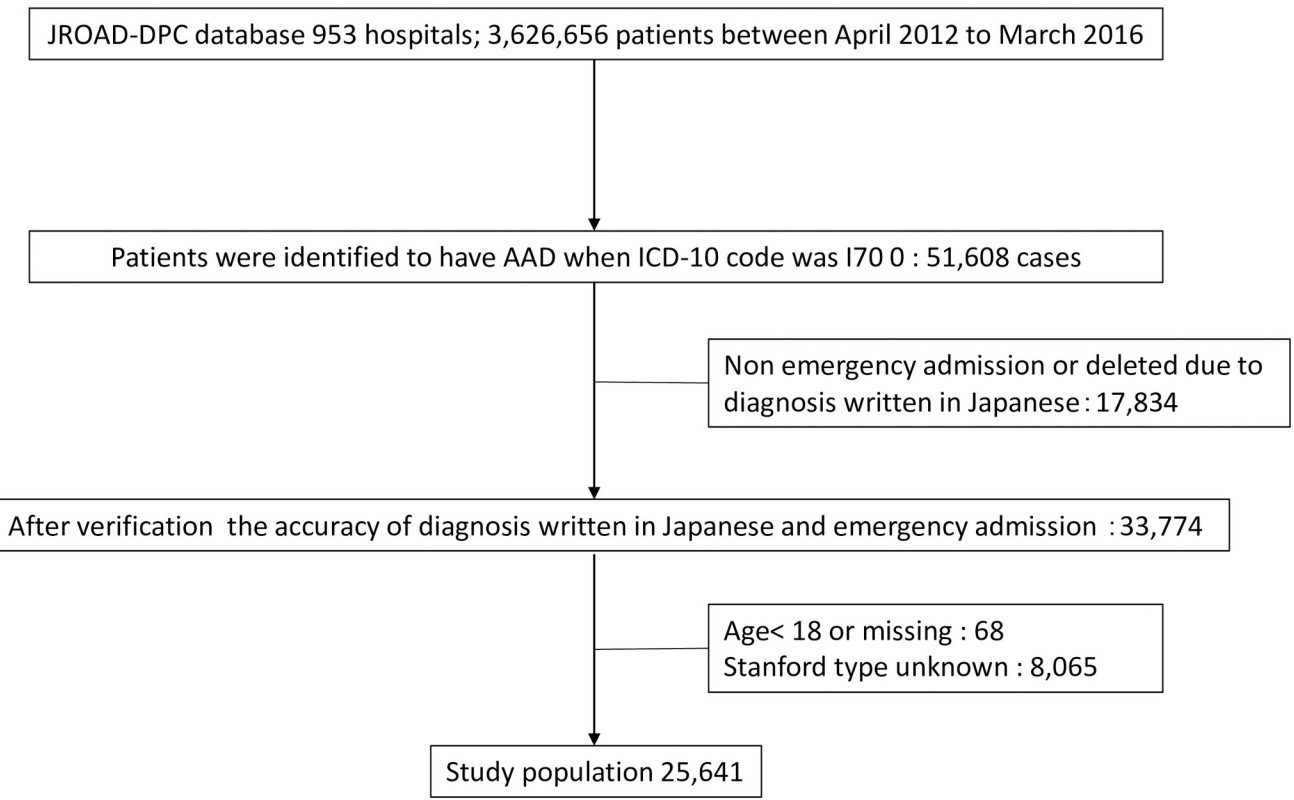

**Fig 1. Flow chart of the patient selection process.** JROAD-DPC, Japanese Registry of all Cardiac and Vascular Diseases Diagnostic Procedure Combination; ICD-10, 10th revision of the International Classification of Disease.

analysis was performed using STATA software, version 16.0 (StataCorp, College Station, Texas, USA). Statistical significance was based on a two-tailed test and defined as a p-value of less than 0.05.

## Results

### Characteristics of the study population

The baseline characteristics of patients according to admission day, Stanford type, and sex are shown in Tables 1 and 2, and S1 Table. Of the 33,706 patients, 56.5% were of the male sex. The mean age ± SD was 67 ±13, and 75 ± 12 years old for men and women, respectively. The prevalence of current smoking was significantly higher (men: 72. 2%, women: 30.4%) than that of the average Japanese population [15]. Type A AAD and Type B AAD were present in 49.9% and 50.1% of patients, respectively. More than half of patients with Type A AAD underwent surgery (58.2%). The in-hospital mortality was 12.8% and 20.4%, in men and women, respectively. In contrast, the in-hospital mortality was 27.6% and 4.5% in patients with Type A and Type B AAD, respectively.

### Weekly variations in the frequency of AAD

Weekly variations in the frequency of AAD admissions are shown in Fig 2. The frequency of AAD admission was significantly greater on weekdays (18.5±7.9, admissions/day, ANOVA p<0.001) than Saturdays (16.3 ±6.9 admissions/day) or Sundays/holidays (15.1 ±6.5 admissions/day) (Fig 2A and 2B).

**Table 1. Characteristics of patients with acute aortic dissection according to admission day.**

| | Weekdays | Saturdays | Sunday/Holidays | p-value |
|---|---|---|---|---|
| Number | 18,301 | 3,268 | 4,072 | |
| Men (%)[a] | 10,486 (57.3) | 1,905 (58.3) | 2,339 (57.4) | 0.57 |
| Age[b] | 70 ± 13 | 70 ± 14 | 70 ± 13 | 0.077 |
| Current smoking (%)[a] | 9,931 (56.5) | 1,783 (54.6) | 2,238 (55.0) | 0.71 |
| Comorbidity | | | | |
| Hypertension | 12,28 (67.9) | 2,203 (67.4) | 2,715 (66.7) | 0.30 |
| Diabetes mellitus | 1,695 (9.3) | 282 (8.6) | 367 (9.0) | 0.49 |
| Dyslipidemia | 3306 (18.1) | 547 (16.7) | 684 (16.8) | 0.049 |
| Heart failure | 3439 (18.8) | 616 (18.8) | 763 (18.7) | 0.99 |
| Myocardial infarction | 644 (3.5) | 111 (3.4) | 131 (3.2) | 0.62 |
| Cerebrovascular disease | 1575 (8.6) | 258 (7.9) | 383 (9.4) | 0.069 |
| Renal disease | 864 (4.7) | 153 (4.7) | 197 (4.8) | 0.94 |
| Cancer | 610 (3.3) | 105 (3.2) | 125 (3.1) | 0.68 |
| Stanford type A | 9076 (49.6) | 1,645 (50.3) | 2,077 (51.0) | 0.23 |
| Surgery (%)[a] | 5,936 (32.4) | 1,064 (32.6) | 1,329 (32.6) | 0.86 |
| Endovascular | 471 (7.9) | 71 (6.7) | 95 (7.1) | 0.27 |
| Open Surgery | 5,465 (92.1) | 993 (93.3) | 1,234 (92.9) | |
| Hospitalization days | 25 ± 21 | 25 ± 20 | 25 ± 22 | 0.002 |
| In-hospital mortality (%)[a] | 2,841 (15.5) | 547 (16.7) | 723 (17.8) | < 0.001 |

[a, b] Data are expressed as number (%), or mean ± standard deviation.

**Table 2. Characteristics of patients with acute aortic dissection according to Stanford type.**

| | Stanford A | Stanford B | p-value |
|---|---|---|---|
| Number | 12,798 | 12,843 | |
| Men (%)[a] | 5,869 (45.9) | 8,861 (69.0) | < 0.001 |
| Age[b] | 71 ± 14 | 70± 13 | 0.001 |
| Current smoking (%)[a] | 6,197 (48.4) | 7,755 (60.4) | < 0.001 |
| Comorbidity | | | |
| Hypertension | 6,970 (54.5) | 10,376 (80.8) | < 0.001 |
| Diabetes mellitus | 1,042 (8.1) | 1,302 (10.1) | < 0.001 |
| Dyslipidemia | 1,471 (11.5) | 3,066 (23.9) | < 0.001 |
| Heart failure | 2,612 (20.4) | 2,206 (17.2) | < 0.001 |
| Myocardial infarction | 542 (4.2) | 344 (2.7) | < 0.001 |
| Cerebrovascular disease | 1,399 (10.9) | 817 (6.4) | < 0.001 |
| Renal disease | 477 (3.7) | 737 (5.7) | < 0.001 |
| Cancer | 327 (2.6) | 513 (4.0) | < 0.001 |
| Surgery (%)[a] | 7,453 (58.2) | 876 (6.8) | < 0.001 |
| Endovascular | 88 (1.2) | 549 (62.7) | < 0.001 |
| Open Surgery | 7,365 (98.8) | 327 (37.3) | |
| Hospitalization days | 25 ± 24 | 24 ± 16 | < 0.001 |
| In-hospital mortality (%)[a] | 3,528 (27.6) | 583 (4.5) | < 0.001 |

[a, b] Data are expressed as number (%), or mean ± standard deviation.

A
ANOVA p<0.001

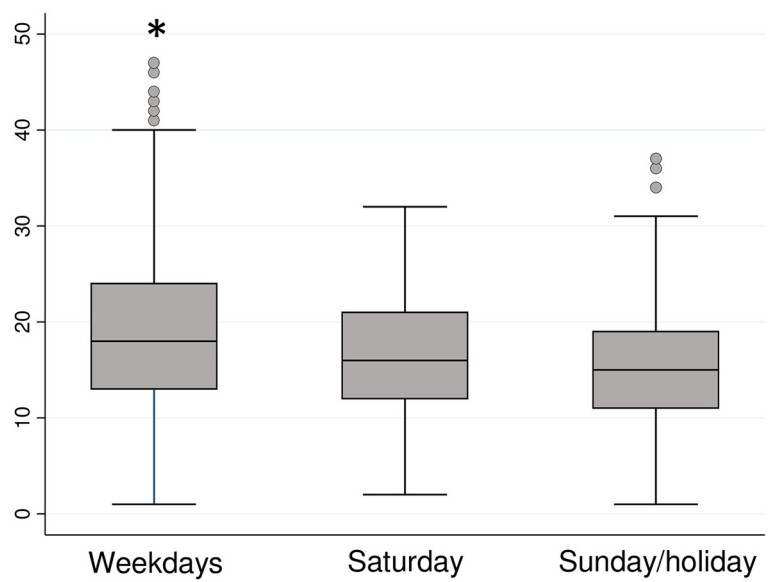

B
ANOVA p<0.001

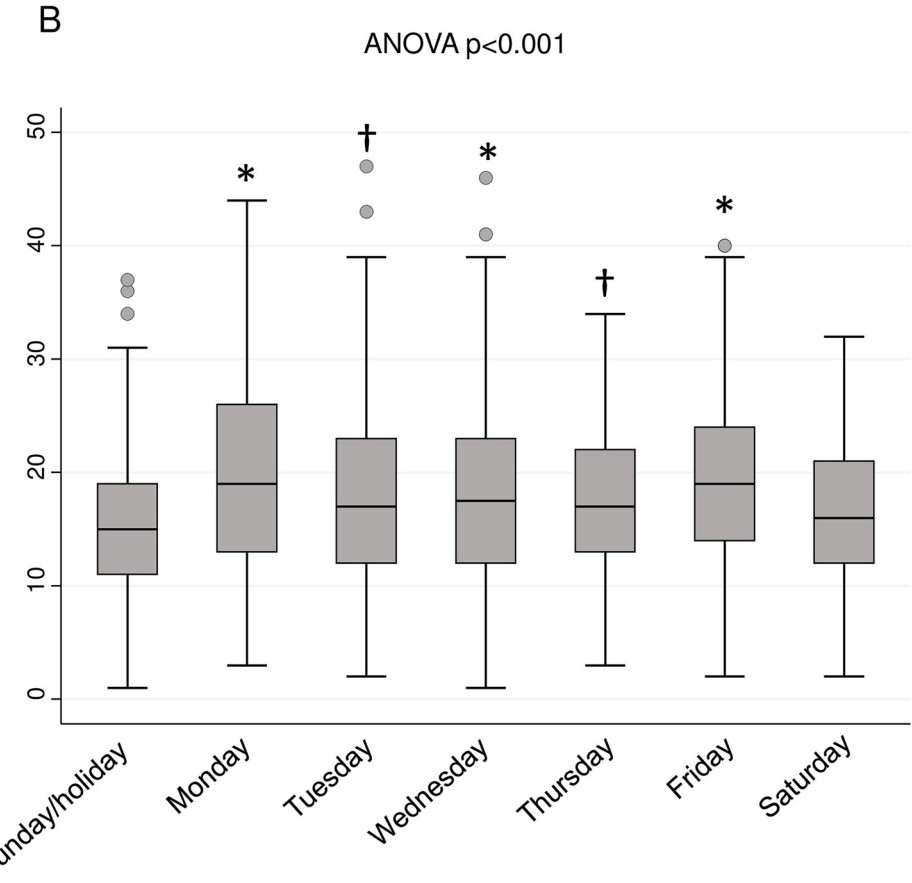

**Fig 2.** Frequency of admission of acute aortic dissection during weekdays, Saturdays, or Sunday/holiday (A), seven days of the week (B). The vertical line shows the average number of admitted patients per day. ANOVA, analysis of variance with post hoc Bonferroni correction. * p <0.001 vs. Sunday/holiday, † p <0.05 vs. Sunday/holiday.

### Logistic regression analyses

After adjusting for sex, age, Stanford type, and surgery, patients admitted on Sundays/holidays were shown to have an increased risk of in-hospital mortality (odds ratio (OR) 1.20; 95% confidence interval [CI] 1.07–1.33, p = 0.001) compared with patients admitted on weekdays (reference) (Table 3). Patients in the Stanford type A AAD group admitted on Sundays/holidays showed a significantly increased risk of in-hospital mortality (non-surgery vs. surgery groups: 95% CI 1.06–1.48 vs. 1.17–1.68; p<0.001 for both groups, OR 1.25 vs. 1.41, respectively) compared with those who were admitted on weekdays. Conversely, those in the Stanford Type B AAD group admitted on Sundays/holidays did not show an increased risk of in-hospital mortality (non-surgery vs. surgery: 95% CI 0.64–1.09 vs. 0.40–2. 10; p = 0.182 vs. 0.846; OR 0.84 vs. 0.92). Patients who were admitted with AAD on Saturdays did not show an increased risk of in-hospital mortality (OR 1.12; 95% CI 0.99–1.25, p = 0.066).

In the same manner, analyses of the seven days group also showed higher in-hospital mortality on Sunday/holiday admissions than weekday admissions (Table 4). In the stratified analysis, a higher in-hospital mortality on Sunday/holiday admissions of Type A AAD was observed, while there was no significant difference in the in-hospital mortality between Sundays/holidays and weekdays in Type B AAD.

## Discussion

This large Japanese nationwide study revealed that the prevalence of AAD admission is more frequent on weekdays Furthermore, those admitted on Sundays/holidays and classified as Type A AAD had a significantly increased risk of in-hospital mortality.

The findings of the present study are congruous with those of Kumar et al. and Gallerani et al. This study also demonstrated that patients with Type A AAD rather than Type B AAD

**Table 3. In-hospital mortality per weekday, Saturday, Sunday/holiday.**

| | | Odds ratio | 95% CI | P |
|---|---|---|---|---|
| Total | Weekdays | 1.00 (reference) | | |
| N = 25,641 | Saturday | 1.12 | 0.99–1.25 | 0.066 |
| | Sunday/holiday | 1.20 | 1.07–1.33 | 0.001 |
| Stanford A | Weekdays | 1.00 (reference) | | |
| non-surgery | Saturday | 1.18 | 0.99–1.41 | 0.072 |
| N = 5,435 | Sunday/holiday | 1.25 | 1.06–1.48 | 0.007 |
| Stanford A | Weekdays | 1.00 (reference) | | |
| surgery | Saturday | 1.03 | 0.83–1.28 | 0.787 |
| N = 7,453 | Sunday/holiday | 1.41 | 1.17–1.68 | < 0.001 |
| Stanford B | Weekdays | 1.00 (reference) | | |
| non-surgery | Saturday | 1.13 | 0.87–1.47 | 0.358 |
| N = 11,967 | Sunday/holiday | 0.84 | 0.64–1.09 | 0.182 |
| Stanford B | Weekdays | 1.00 (reference) | | |
| surgery | Saturday | 0.98 | 0.41–2.34 | 0.959 |
| N = 876 | Sunday/holiday | 0.92 | 0.40–2.10 | 0.846 |

CI, confidence interval.

**Table 4. In-hospital mortality per 7 days.**

| | | Odds ratio | 95% CI | P |
|---|---|---|---|---|
| Total N = 25,641 | Sunday/holiday | 1.00 (reference) | | |
| | Monday | 0.83 | 0.72–0.96 | 0.010 |
| | Tuesday | 0.88 | 0.77–1.01 | 0.071 |
| | Wednesday | 0.78 | 0.68–0.90 | 0.001 |
| | Thursday | 0.97 | 0.84–1.11 | 0.623 |
| | Friday | 0.74 | 0.64–0.85 | <0.001 |
| | Saturday | 0.93 | 0.81–1.08 | 0.338 |
| Stanford A non-surgery N = 5,345 | Sunday/holiday | 1.00 (reference) | | |
| | Monday | 0.76 | 0.61–0.95 | 0.014 |
| | Tuesday | 0.86 | 0.69–1.07 | 0.176 |
| | Wednesday | 0.81 | 0.65–1.00 | 0.055 |
| | Thursday | 0.91 | 0.73–1.14 | 0.422 |
| | Friday | 0.69 | 0.55–0.85 | 0.001 |
| | Saturday | 0.94 | 0.75–1.18 | 0.605 |
| Stanford A surgery N = 7,453 | Sunday/holiday | 1.00 (reference) | | |
| | Monday | 0.78 | 0.61–1.00 | 0.054 |
| | Tuesday | 0.72 | 0.56–0.93 | 0.010 |
| | Wednesday | 0.62 | 0.48–0.81 | <0.001 |
| | Thursday | 0.81 | 0.64–1.04 | 0.099 |
| | Friday | 0.63 | 0.49–0.81 | <0.001 |
| | Saturday | 0.73 | 0.57–0.95 | 0.017 |
| Stanford B non-surgery N = 11,967 | Sunday/holiday | 1.00 (reference) | | |
| | Monday | 1.12 | 0.80–1.58 | 0.505 |
| | Tuesday | 1.26 | 0.90–1.76 | 0.180 |
| | Wednesday | 1.11 | 0.79–1.57 | 0.543 |
| | Thursday | 1.37 | 0.99–1.90 | 0.058 |
| | Friday | 1.11 | 0.80–1.56 | 0.529 |
| | Saturday | 1.35 | 0.96–1.90 | 0.081 |
| Stanford B surgery N = 876 | Sunday/holiday | 1.00 (reference) | | |
| | Monday | 1.05 | 0.37–2.97 | 0.922 |
| | Tuesday | 1.56 | 0.59–4.12 | 0.372 |
| | Wednesday | 0.43 | 0.12–1.53 | 0.193 |
| | Thursday | 1.28 | 0.46–3.55 | 0.637 |
| | Friday | 1.20 | 0.42–3.40 | 0.736 |
| | Saturday | 1.05 | 0.35–3.19 | 0.925 |

CI, confidence interval.

patients, admitted on a Sunday/holiday showed a significantly increased risk of in-hospital mortality when compared with weekday admissions. In contrast, the "weekend effect" was not observed in studies by Gokalp et al. and Ahlsson et al. The reasons for the differences in results remain unclear. However, the differences may be attributable to the relatively small sample sizes of the two studies (Gokalp et al., 206 patients; Ahlsson et al., 1,159 patients) when compared with that of the present study. Moreover, medical systems differ across regions and different time periods; thus, considering the results of the present study, the "weekend effect" may have become apparent in recent years in patients with Type A AAD, in Japan.

There have been speculations regarding the reasons for the occurrence of the "weekend effect." Firstly, the "weekend effect" is partly responsible for the number of severely ill patients

admitted to hospitals during the weekends [16, 17]. Sun et al. reported that the "weekend effect" was not significant if adjusted by illness severity [16]. The reason for the increased numbers of severely affected patients admitted during weekends may be the many referrals from general practitioners and direct admission to teaching hospitals during weekends. This study did not adjust for severity of illness except for the Stanford Type; however, weekend admissions had a smaller proportion of referred patients than those admitted on weekdays [44.1%, 39.3%, 35.2% on weekdays, Saturdays, Sundays/holidays, respectively]. It is conceivable that many patients with AAD were admitted directly to hospitals after the disease onset; therefore, a small selection bias may exist.

Secondly, differences in the staffing and resources of hospitals between weekdays and weekends need to be considered. Fewer people work in hospitals during the weekends/holidays than on weekdays. Inadequate staffing resources [18] and staffing levels [19] may have influenced the increase in mortality. In addition, weekend on-call teams of surgeons may have less expertise and experience in working together as compared to the weekday teams [4]. Moreover, many doctors who work on weekends may have had an excessive workload, which in turn, may have an influence on mortality rates. Fatigue in medical teams fatigue is associated with worse patient outcomes [20]. Further studies should explore the reasons underlying the "weekend effect" on AAD in-hospital mortality.

The results of the present study have shown that only patients with Type A AAD who were admitted on a Sunday/holiday, especially the surgery group, showed a significantly increased risk of in-hospital mortality when compared with weekday admissions. We speculate that the differences between Type A and Type B AAD are due to the lack of complications in Type B AAD. In the absence of malperfusion or signs of (early) disease progression, these patients can be stabilized safely using medical therapy alone to control pain and blood pressure [11], whereas in patients with Type A AAD, surgery is the treatment choice. Therefore, Type B AAD seems to be less influenced by the differences in staff or resources between weekday and weekend admissions.

This study has several limitations. First, this was a comparative analysis of observational data; therefore, risk of misdiagnosis may exist despite our careful selection of patients and surgical procedures. Second, we did not have detailed data regarding the precise time of onset of AAD, and thus we could not analyze circadian patterns. We also did not have individual information regarding the disease severity, laboratory data, presence of comorbidities, past medical history, family history, socioeconomic status, and in-hospital mortality cause. Third, as mentioned previously, the quality of medical-care systems and available resources, as well as the "weekend effect," may differ between countries; therefore, caution should be exercised before extrapolating our results to other countries.

Despite these limitations, our study provides noteworthy information regarding the "weekend effect," as data of several patients with AAD have not been previously analyzed. In conclusion, this Japanese nationwide observational study showed that there are more frequent AAD admissions during weekdays. Moreover, patients with Stanford Type A AAD admitted on a Sunday/holiday have an increased risk of in-hospital mortality. Further studies are required to investigate whether uniform diagnosis and therapeutic models used throughout the week can prevent worse outcomes in weekend- admission patients.

## Supporting information

**S1 Table. Characteristics of patients with acute aortic dissection according to sex.**
(DOCX)

## Acknowledgments

The authors thank Dr. Koichi Akutsu for his valuable suggestions for improving the manuscript.

## Author Contributions

**Conceptualization:** Katsuhito Kato, Toshiaki Otsuka, Yoshihiko Seino, Tomoyuki Kawada.

**Data curation:** Katsuhito Kato, Michikazu Nakai, Yoko Sumita.

**Formal analysis:** Michikazu Nakai.

**Funding acquisition:** Katsuhito Kato.

**Investigation:** Tomoyuki Kawada.

**Methodology:** Katsuhito Kato, Toshiaki Otsuka, Michikazu Nakai.

**Project administration:** Katsuhito Kato, Yoshihiko Seino, Tomoyuki Kawada.

**Resources:** Katsuhito Kato.

**Software:** Katsuhito Kato, Michikazu Nakai, Yoko Sumita.

**Supervision:** Toshiaki Otsuka, Tomoyuki Kawada.

**Validation:** Toshiaki Otsuka, Michikazu Nakai, Yoko Sumita, Yoshihiko Seino, Tomoyuki Kawada.

**Writing – original draft:** Katsuhito Kato, Toshiaki Otsuka.

**Writing – review & editing:** Michikazu Nakai, Yoko Sumita, Yoshihiko Seino, Tomoyuki Kawada.

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
