## [Decision Letter · Decision Letter 0]

6 Jul 2021

PONE-D-21-14016

Effect of weekend admission for acute aortic dissection on in-hospital mortality in Japan : a nationwide study

PLOS ONE

Dear Dr. Kato

Thank you for submitting your manuscript to PLOS ONE. After careful consideration, even if the manuscript describes an original topic, we feel that it has merit but does not fully meet PLOS ONE’s publication criteria as it currently stands. Therefore, we invite you to submit a revised version of the manuscript that addresses the points raised during the review process.

We look forward to receiving your revised manuscript.

Kind regards,

Alessandro Leone, MD

Academic Editor

PLOS ONE

Journal Requirements:

Reviewers' comments:

Reviewer's Responses to Questions

**Comments to the Author**

1. Is the manuscript technically sound, and do the data support the conclusions?

Reviewer #1: Partly

Reviewer #2: Yes

Reviewer #3: Partly

Reviewer #4: Partly

2. Has the statistical analysis been performed appropriately and rigorously? 

Reviewer #1: No

Reviewer #2: No

Reviewer #3: Yes

Reviewer #4: No

3. Have the authors made all data underlying the findings in their manuscript fully available?

Reviewer #1: Yes

Reviewer #2: No

Reviewer #3: Yes

Reviewer #4: Yes

4. Is the manuscript presented in an intelligible fashion and written in standard English?

Reviewer #1: Yes

Reviewer #2: Yes

Reviewer #3: Yes

Reviewer #4: Yes

5. Review Comments to the Author

Reviewer #1: This is a retrospective observational study in which 33706 patients admitted for acute aortic dissection in 953 certified Japanese hospital were analysed in order to find a correlation between admission day and in-hospital mortality. The patients were categorized according to the day of hospital admission and the analysis was performed comparing the patients that were admitted in the hospital and treated (surgically or medically) on weekdays, Saturdays and Sundays/Holidays. Thereafter, the analysis was carried out by each day of the week. The study showed that the patients with acute type A aortic dissection admitted on Sunday/holiday had a significantly increased risk of in-hospital mortality. In contrast, the patients admitted for acute type B aortic dissection didn’t show a difference in terms of in-hospital mortality according to the day of admission to the hospital.

The topic of this study is very interesting and the potentialities of the analysis, including a large cohort of patients, are higher.

The manuscript is well written, with proper English.

However, there are some points of discussion:

1. The abstract should be divided in the appropriate sections.

2. The method of categorization of patients used to perform the analysis was described in the section “Data definition”. It should be moved in the section “Study setting and patients selection”.

3. In the section “Measurement”, it was described only the primary outcome of the study. This item should be improved.

4. The information regarding the patient’s selection and the definition of population analysed in the study, in association with the flow-chart of the patient selection process, were reported in the section “Results”. It should be moved in the “Methods” section.

5. The Table 1 and Table 2 showed the characteristics of the patients according to sex and to Stanford type. The variables analysed are few. In particular, many pre-operative variables, including cardiovascular risk factors, that could help to understand the risk profile of patients are not analysed.

6. In Table 2, they reported that 8065 patients (23.9%) of the overall population were “Unknown”. This group of patients should be removed from the analysis.

7. In Table 2, the authors reported that 58.2% of acute type A aortic dissection were treated surgically. How were the remaining patients treated?

8. The type of aortic surgery performed is not reported.

9. The post-operative complications of patients underwent aortic arch surgery should be added.

10. The authors reported in the limitation of the study the lack of the pre-operative, intra-operative and post-operative information, but these data are important to comprehend the risk profile of the patients and the results of the study.

11. The Table 3 reported the logistic regression analysis for in-hospital mortality for weekday, Saturday, Sunday/holiday and for every 7 days of the week. The two different analysis are reported in two different tables that should be numbered separately. Moreover, in these tables the results are difficult to understand.

12. Among acute type B aortic dissection, both surgically and medically treated patients were included. These two types of patient population could not be considered in the same group. The patients with acute type B aortic dissection surgically treated have anatomic and hemodynamic instability requiring urgent surgical/endovascular treatment. Instead, patients medically treated are stable and had lower risk of morbidity and mortality.

13. In order to find a correlation between admission day and in-hospital mortality, it may be more interesting to analyse in detail the single population of patients admitted in the hospital for acute type A aortic dissection and treated with emergent surgical operation. In fact, as the authors of this study also report, the acute Type A aortic dissection seems more vulnerable to “weekend effect” than the acute Type B aortic dissection medically treated.

Reviewer #2: Congratulations for selecting one of the most important diseases in CTS.

Methods

As far as I can tell, you have used ANOVA for comparing weekdays vs weekends.

Have you performed ad hoc analysis in order to determine where the significant results peak? Or pairwise testing with P-value correction.

You use frequently stratified analysis term: make describe it under statistical analysis

Results

Table 2 lacks a statistical test between Type A and B.

In both logistic regression analysis your weekdays vs sunday/holiday is containing saturday?

In stratified analysis p-value and CI reporting there is discordance: if CI contains 1 results can not be significant, please do revisit your analysis

From table make clear comparing groups with its p values

I truly and honestly think there is something else, only week day can not explain the complexity of the AAD mortality.

It could be interesting to understand time of admission (morning shift or evening shift), shortage of professionals, ect.

Please do share your depersonalized data.

Reviewer #3: 1. Aortic dissection type-A and type B have very different severities and they are very distinct disease entities: most A need big open surgery and most B need only endovascular or non-surgical treatments. They should not be mixed in the same article. If it does, please make sure to discuss them separately.

2. Here the grouping was defined as "admission date". Please specify in more detail. Was it referred to actual clock time or admin time? Was your date starting from 8:01 am or 0:01 am? For example, was 1 am on Monday classified as Monday or Sunday? By admin time, Monday starts from 8am Monday but manpower at 1 am Monday still belongs to Sunday. Please clarify. This confusion will cause a huge misclassification.

3. Here in the Title "weekends" always include Saturdays. But in your manuscript, Saturdays were not treated the same as Sunday or holidays. If your major grouping was Sundays/holidays vs weekdays/non-holidays, please revise your Title accordingly--it's not "weekend" effect! Revised to "holiday" effect or maybe some other better term. Please clarify.

4. What are the non-weekend holidays in Japan? Please specify. Also, was there any holidays celebrated differently in different parts in Japan--namely, was there any holiday off in some parts but non-off in other parts in Japan? In this nationwide study, this geographical holiday variations must be considered if any.

5. The outcome of "in-hospital mortality" needs more clarification. In Asian culture, some family prefer the patients to die at home What if the dying patient, alive when leaving hospital, was transferred home to die naturally. Was it "in-hospital mortality"? Also, how about the OHCA cases? If they were resuscitated and able to be diagnosed as aortic dissection, were they included or excluded in the study cohort? If these OHCA aortic dissection cases were supported by ECMO throughout the course until withdrawal, were they (dead before arrival to hospital) considered "in-hospital death"?

There are some confusions requiring clarification in the manuscript. Please address them properly.

Reviewer #4: the study sounds interesting but difference between groups (weekend and weekday) are missing and the final results should be adjuasted for differences. Table 1 reports difference according to gender which is not the issue under evaluation. Without a correct adjustment no conclusions can be traced

6. PLOS authors have the option to publish the peer review history of their article (what does this mean?). If published, this will include your full peer review and any attached files.

Reviewer #1: No

Reviewer #2: No

Reviewer #3: **Yes: **Robert J. Chen, MD, MPH

Reviewer #4: No

---

## [Author Response · Author response to Decision Letter 0]

15 Sep 2021

RESPONSE TO THE REVIEWERS

On behalf of all the authors, I would like to thank the Reviewers for their valuable suggestions, which have helped us to revise and improve our manuscript. I would also like to thank the Editor for the kind remarks.

RESPONSE TO REVIEWER:

Reviewer #1: 

1. The abstract should be divided in the appropriate sections.

Thank you for pointing this out. We have divided the Abstract sections: Background, Methods, Results, and Conclusion.

2. The method of categorization of patients used to perform the analysis was described in the section “Data definition”. It should be moved in the section “Study setting and patients selection”.

Thank you for pointing this out. We have moved these sentences to the “Study setting and patients selection” section.

3. In the section “Measurement”, it was described only the primary outcome of the study. This item should be improved.

Based on your comment, we have added information on the secondary outcome and variables that may be controversial, such as in-hospital mortality.

4. The information regarding the patient’s selection and the definition of population analysed in the study, in association with the flow-chart of the patient selection process, were reported in the section “Results”. It should be moved in the “Methods” section.

Based on your comment, we have moved the said sentences to the “patients’ selection process” in the Methods.

5. The Table 1 and Table 2 showed the characteristics of the patients according to sex and to Stanford type. The variables analysed are few. In particular, many pre-operative variables, including cardiovascular risk factors, that could help to understand the risk profile of patients are not analysed.

Thank you for pointing this out. We have added data of patients’ comorbidities (i.e. hypertension, diabetes, dyslipidemia, congestive heart failure, myocardial infarction, cerebrovascular disease, renal disease, and cancer). 

6. In Table 2, they reported that 8065 patients (23.9%) of the overall population were “Unknown”. This group of patients should be removed from the analysis.

We have removed the “Unknown” group of Stanford type patients from the analysis. As a result, we reanalyzed our data and reported the results accordingly.

7. In Table 2, the authors reported that 58.2% of acute type A aortic dissection were treated surgically. How were the remaining patients treated?

Within this database, we could not know the detailed treatment progression of each patient. The remaining patients were medically treated.

8. The type of aortic surgery performed is not reported.

We have added the type of surgical intervention (endovascular or open surgery) in the Tables. Open surgery was performed in 98.8% of the Stanford type A group. On the other hand, endovascular repair was performed in 62.7% of the Stanford type B group.

9. The post-operative complications of patients underwent aortic arch surgery should be added.

Unfortunately, this database does not contain information about post-operative complications. We mentioned this as one of the study limitations (lack of postoperative date in the database).

10. The authors reported in the limitation of the study the lack of the pre-operative, intra-operative and post-operative information, but these data are important to comprehend the risk profile of the patients and the results of the study.

We acknowledge the importance of this information in this study. However, the database does not contain information about post-operative complications.

11. The Table 3 reported the logistic regression analysis for in-hospital mortality for weekday, Saturday, Sunday/holiday and for every 7 days of the week. The two different analysis are reported in two different tables that should be numbered separately. Moreover, in these tables the results are difficult to understand.

Based on your comment, we have separated these two tables (Table 3 and Table 4).

12. Among acute type B aortic dissection, both surgically and medically treated patients were included. These two types of patient population could not be considered in the same group. The patients with acute type B aortic dissection surgically treated have anatomic and hemodynamic instability requiring urgent surgical/endovascular treatment. Instead, patients medically treated are stable and had lower risk of morbidity and mortality.

Based on your suggestion, we divided the patients into the four groups (Stanford type A and surgical intervention (-), Stanford type A and surgical intervention (+), Stanford type B and surgical intervention (-), and Stanford type B and surgical intervention (+)) in the stratified analysis. In the present analysis, only 6.8% of patients with Stanford type B had surgical intervention. In the stratified analysis, in-hospital mortality of patients with Stanford type B admitted on　Sunday/holiday in both the surgical intervention (-) and (+) group was not significantly different from those admitted during the weekdays. 

13. In order to find a correlation between admission day and in-hospital mortality, it may be more interesting to analyse in detail the single population of patients admitted in the hospital for acute type A aortic dissection and treated with emergent surgical operation. In fact, as the authors of this study also report, the acute Type A aortic dissection seems more vulnerable to “weekend effect” than the acute Type B aortic dissection medically treated.

Based on your suggestion, we divided patients with Stanford type A according to presence or absence of surgical intervention. In the stratified analysis, patients with Stanford type A and presence of surgical intervention seemed more vulnerable to the “weekend effect” than absence of surgical intervention. 

Reviewer #2: 

Methods

14. As far as I can tell, you have used ANOVA for comparing weekdays vs weekends.　Have you performed ad hoc analysis in order to determine where the significant results peak? Or pairwise testing with P-value correction.

Given that this study did not aim at knowing the differences between the particular days, we did not perform an ad hoc analysis. This study rather focused on investigating the difference between the presence or absence. 

15. You use frequently stratified analysis term: make describe it under statistical analysis

Thank you for pointing this out. We have improved the statistical analysis section.

Results

16. Table 2 lacks a statistical test between Type A and B.

Based on your comment, we added the statistical test used in comparing between Type A and B groups.

17. In both logistic regression analysis your weekdays vs sunday/holiday is containing saturday?

Saturday was included in both logistic analyses. 

18. In stratified analysis p-value and CI reporting there is discordance: if CI contains 1 results can not be significant, please do revisit your analysis

Based on your suggestion, we performed a re-analysis and provided the CIs where available, to justify the significant differences.

19. From table make clear comparing groups with its p values

Thank you for pointing this out. We have included the p values in the tables and added more information on the statistical test used in the Method.

20. I truly and honestly think there is something else, only week day can not explain the complexity of the AAD mortality. It could be interesting to understand time of admission (morning shift or evening shift), shortage of professionals, ect. Please do share your depersonalized data.

As your righty commented, the factors influening AAD mortality are complex. The weekend effect involves several factors; however, our database did not contain the exact time of admission. 

Reviewer #3: 

21. Aortic dissection type-A and type B have very different severities and they are very distinct disease entities: most A need big open surgery and most B need only endovascular or non-surgical treatments. They should not be mixed in the same article. If it does, please make sure to discuss them separately.

Thank you for pointing this out. We stratified our data according to the presence/absence of surgical intervention and presented the results accordingly. Furthermore, we discussed the findings in the Discussion section.

22. Here the grouping was defined as "admission date". Please specify in more detail. Was it referred to actual clock time or admin time? Was your date starting from 8:01 am or 0:01 am? For example, was 1 am on Monday classified as Monday or Sunday? By admin time, Monday starts from 8am Monday but manpower at 1 am Monday still belongs to Sunday. Please clarify. This confusion will cause a huge misclassification.

Thank you for pointing this out. The time range for admissions in our study was 00:00 am–11:59 pm. 　The patients’ real admission time reflected the admission day. 

23. Here in the Title "weekends" always include Saturdays. But in your manuscript, Saturdays were not treated the same as Sunday or holidays. If your major grouping was Sundays/holidays vs weekdays/non-holidays, please revise your Title accordingly--it's not "weekend" effect! Revised to "holiday" effect or maybe some other better term. Please clarify.

Thank you for pointing this out. We have changed the title from “weekend admission” to “holiday admission.”

24. What are the non-weekend holidays in Japan? Please specify. Also, was there any holidays celebrated differently in different parts in Japan--namely, was there any holiday off in some parts but non-off in other parts in Japan? In this nationwide study, this geographical holiday variations must be considered if any.

Non-weekend holidays (i.e., national holidays) are same in all parts of Japan. We have added information in the methods to clarify this issue.

25. The outcome of "in-hospital mortality" needs more clarification. In Asian culture, some family prefer the patients to die at home What if the dying patient, alive when leaving hospital, was transferred home to die naturally. Was it "in-hospital mortality"? Also, how about the OHCA cases? If they were resuscitated and able to be diagnosed as aortic dissection, were they included or excluded in the study cohort? If these OHCA aortic dissection cases were supported by ECMO throughout the course until withdrawal, were they (dead before arrival to hospital) considered "in-hospital death"? There are some confusions requiring clarification in the manuscript. Please address them properly.

Thank you for pointing these out. Patients who were in a dying state upon discharge (left to die naturally at home) or patients supported by ECMO throughout the course until withdrawal were not considered as “in-hospital mortality.” It is true that, some patients and their families in Japan prefer to die at home, however, very few patients in dying state with emergency conditions were left to go home. Furthermore, our database contained data of patients who were brought to the hospital for cardiac arrest state and were diagnosed with AAD. Such patients’ deaths were considered in-hospital mortality.

Reviewer #4: 

26. Difference between groups (weekend and weekday) are missing and the final results should be adjuasted for differences. Table 1 reports difference according to gender which is not the issue under evaluation. Without a correct adjustment no conclusions can be traced

We have changed Table 1 from the difference between men and women to difference between weekend and weekday groups. The table according to sex has been considered as Supplementary Table1.

---

## [Decision Letter · Decision Letter 1]

11 Oct 2021

PONE-D-21-14016R1Effect of holiday admission for acute aortic dissection on in-hospital mortality in Japan : a nationwide studyPLOS ONE

Dear Dr. Kato

Thank you for submitting your manuscript to PLOS ONE. After careful consideration, we feel that it has merit but does not fully meet PLOS ONE’s publication criteria as it currently stands. Therefore, we invite you to submit a revised version of the manuscript that addresses the points raised during the review process.

We look forward to receiving your revised manuscript.

Kind regards,

Alessandro Leone, MD

Academic Editor

PLOS ONE

Journal Requirements:

Additional Editor Comments (if provided):

Dear Author, in my opinion the manuscript presents all the criteria for a publication, however before proceeding I kindly ask you to address the last and minor issue pointed out by Reviewer 2.

Reviewers' comments:

Reviewer's Responses to Questions

**Comments to the Author**

1. If the authors have adequately addressed your comments raised in a previous round of review and you feel that this manuscript is now acceptable for publication, you may indicate that here to bypass the “Comments to the Author” section, enter your conflict of interest statement in the “Confidential to Editor” section, and submit your "Accept" recommendation.

Reviewer #1: All comments have been addressed

Reviewer #2: All comments have been addressed

Reviewer #3: All comments have been addressed

Reviewer #4: All comments have been addressed

2. Is the manuscript technically sound, and do the data support the conclusions?

Reviewer #1: Yes

Reviewer #2: Yes

Reviewer #3: Yes

Reviewer #4: Yes

3. Has the statistical analysis been performed appropriately and rigorously? 

Reviewer #1: Yes

Reviewer #2: Yes

Reviewer #3: Yes

Reviewer #4: Yes

4. Have the authors made all data underlying the findings in their manuscript fully available?

Reviewer #1: Yes

Reviewer #2: No

Reviewer #3: Yes

Reviewer #4: Yes

5. Is the manuscript presented in an intelligible fashion and written in standard English?

Reviewer #1: Yes

Reviewer #2: Yes

Reviewer #3: Yes

Reviewer #4: Yes

6. Review Comments to the Author

Reviewer #1: The authors modified the paper based on the suggested changes.

In particular:

1. The abstract was divided in the appropriate sections.

2. The information regarding the patient’s selection and the definition of population analysed in the study were included in the appropriate paragraph.

3. In the section “Measurement”, it was described exhaustively all the outcomes of the study.

4. In the Table 1 and Table 2 the pre-operative variables, including cardiovascular risk factors, were added.

5. The 8065 patients (23.9%) included in the “Unknown” section were removed from the analysis.

6. The type of aortic surgery performed and the post-operative complications of patients underwent aortic arch surgery were not added to the analysis. However, the authors explain that these types of information were not available in the database and this problem was mentioned as one of the study limitations.

7. The authors divided the patients into four groups (Stanford type A and surgical intervention (-), Stanford type A and surgical intervention (+), Stanford type B and surgical intervention (-), and Stanford type B and surgical intervention (+)) in the stratified analysis, as suggested. Thus, the readers could better understand which kind of patient’s population is more vulnerable to the “weekend effect”.

Reviewer #2: Methods

Not clear what stands for surgical intervention (-) or (+), make clearer statements.

Results

In all tables in caption or footnote the specific test has been used for comparison.

In Figure 2 try to use different types of plot, for example boxplot! And provide an ad hoc test for better weekday separation and significance.

Reviewer #3: The revision addressed the concerns and issues I uncovered in my previous reviews. The issues of time definition and potential selection bias were addressed or acknowledged.

Reviewer #4: I have no more comment. The authors adequately addressed all the comments. The manuscript is now improved

7. PLOS authors have the option to publish the peer review history of their article (what does this mean?). If published, this will include your full peer review and any attached files.

Reviewer #1: **Yes: **Marianna Berardi

Reviewer #2: **Yes: **Rafik Margaryan MD, PhD

Reviewer #3: **Yes: **Robert Jeenchen Chen, MD, MPH

Reviewer #4: No

---

## [Author Response · Author response to Decision Letter 1]

26 Oct 2021

RESPONSE TO THE REVIEWERS

On behalf of all the authors, I would like to thank the Reviewers for their valuable suggestions, which have helped us to revise and improve our manuscript. I would also like to thank the Editor for the kind remarks.

RESPONSE TO REVIEWER:

Reviewer #2: Methods

Not clear what stands for surgical intervention (-) or (+), make clearer statements.

Response: Thank you for raising this point. We have revised the　descriptions to ensure clarity and avoid any confusion.

Results

In all tables in caption or footnote the specific test has been used for comparison.

In Figure 2 try to use different types of plot, for example boxplot! And provide an ad hoc test for better weekday separation and significance.

Response: Thank you for your suggestion. We have changed Figure 2 to a boxplot and performed an additional ad hoc test.

---

## [Editor Report · Decision Letter 2]

4 Nov 2021

Effect of holiday admission for acute aortic dissection on in-hospital mortality in Japan : a nationwide study

PONE-D-21-14016R2

Dear Dr. Kato,

We’re pleased to inform you that your manuscript has been judged scientifically suitable for publication and will be formally accepted for publication once it meets all outstanding technical requirements.

Kind regards,

Alessandro Leone, MD

Academic Editor

PLOS ONE

Additional Editor Comments (optional):

Dear Authors, Im pleased to inform you that the manuscript, after final revision, can be finally accepted.
---

## [Editor Report · Acceptance letter]

9 Nov 2021

PONE-D-21-14016R2 

Effect of holiday admission for acute aortic dissection on in-hospital mortality in Japan: a nationwide study 

Dear Dr. Kato:

I'm pleased to inform you that your manuscript has been deemed suitable for publication in PLOS ONE. Congratulations! Your manuscript is now with our production department. 

Kind regards, 

on behalf of

Dr. Alessandro Leone 

Academic Editor

PLOS ONE